# Label Embedding via Low-Coherence Matrices

**Jianxin Zhang**
**Electrical Engineering and Computer Science**
**University of Michigan**
**Ann Arbor, MI 48109**
`jianxinz@umich.edu`

**Clayton Scott**
**Electrical Engineering and Computer Science**
**University of Michigan**
**Ann Arbor, MI 48109**
`clayscot@umich.edu`

**Reviewed on OpenReview:** `https://openreview.net/forum?id=vrcWXcr4On`

## Abstract

Label embedding is a framework for multiclass classification problems where each label is represented by a distinct vector of some fixed dimension, and training involves matching model output to the vector representing the correct label. While label embedding has been successfully applied in extreme classification and zero-shot learning, and offers both computational and statistical advantages, its theoretical foundations remain poorly understood. This work presents an analysis of label embedding in the context of extreme multiclass classification, where the number of classes $C$ is very large. We present an excess risk bound that reveals a trade-off between computational and statistical efficiency, quantified via the coherence of the embedding matrix. We further show that under the Massart noise condition, the statistical penalty for label embedding vanishes with sufficiently low coherence. Our analysis supports an algorithm that is simple, scalable, and easily parallelizable, and experimental results demonstrate its superiority in large-scale applications.

## 1 Introduction

In classification, the goal is to learn from feature-label pairs $\{(x_i, y_i)\}_{i=1}^{N}$ a classifier $h : \mathcal{X} \rightarrow \mathcal{Y}$ that accurately maps a feature vector $x$ in the feature space $\mathcal{X}$ to its label $y$ in the label space $\mathcal{Y}$. For implementation purposes, labels are typically represented using the one-hot encoding which, for a $C$-class classification problem, represents the $i$-th class by the $i$-th standard basis vector in $\mathbb{R}^C$.

Unfortunately, the one-hot encoding of labels is limited in the context of *extreme classification*, which refers to multiclass and multilabel classification problems involving thousands of classes or more (Wei et al., 2022). Extreme classification has emerged as an essential research area in machine learning, owing to an increasing number of real-world applications involving massive numbers of classes, such as image recognition (Zhou et al., 2014), natural language processing (Le and Mikolov, 2014; Jernite et al., 2017), and recommendation systems (Bhatia et al., 2015; Chang et al., 2020). Classification methods based on the one-hot encoding often struggle to scale effectively in these scenarios due to the high computational cost and memory requirements associated with handling large label spaces.

To overcome this challenge, the *label embedding* framework represents each label by a vector of fixed dimension, typically much lower than $C$, and learns a function that maps a feature vector to the vector representing its label. At inference time, the label of a test data point is assigned to match the nearest label representative in the embedding space. By employing a lower-dimensional label space, label embedding overcomes the

computational challenge posed by large label space. At the same time, there is a growing need for efficient and scalable algorithms that can tackle extreme classification problems without compromising on performance (Prabhu and Varma, 2014; Prabhu et al., 2018; Deng et al., 2018). Successful applications of label embedding to extreme classification include Yu et al. (2014); Jain et al. (2019); Bhatia et al. (2015); Guo et al. (2019); Evron et al. (2018); Hsu et al. (2009). Furthermore, Rodríguez et al. (2018) argues that label embedding can accelerate the convergence rate and better capture latent relationships between categories.

Despite its widespread use, the theoretical basis of label embedding has not been thoroughly explored. This paper presents a new excess risk bound that provides insight into how label embedding algorithms work, and then exploits this insight to identify a simple implementation of label embedding with excellent performance. Specifically, our bound establishes a trade-off between computational efficiency and classification accuracy, quantified in terms of the coherence of the label embedding matrix. Our theory applies to label-embedding algorithms described by Meta-Algorithm 1, including various types of embeddings: data-independent embeddings (Weston et al., 2002; Hsu et al., 2009), those anchored in auxiliary information (Akata et al., 2013), and embeddings co-trained with models (Weston et al., 2010). Furthermore, under the multiclass noise condition of Massart and Nédélec (2006), the statistical penalty associated with a positive matrix coherence, which results from reducing the dimension of the label space, disappears.

Meta-Algorithm 1 is not our contribution, but rather a unification of several existing approaches. However, by selecting a label embedding matrix to optimize the tradeoff described by our bound, namely, by choosing a low-coherence embedding matrix, we present instantiations of this meta-algorithm that outperform competing approaches across several benchmark datasets.

## 2 Related work

While label embedding has also been successfully applied to zero-shot learning (Wang et al., 2019; Akata et al., 2013), we focus here on extreme classification, together with related theoretical contributions.

### 2.1 Extreme classification

Besides label embedding, existing methods for extreme multiclass classification can be grouped into three main categories: label hierarchy, one-vs-all methods, and other methods.

**Label Embedding.** LEML (Yu et al., 2014) leverages a low-rank assumption on linear models and effectively constrains the output space of models to a low-dimensional space. SLEEC (Bhatia et al., 2015) is a local embedding framework that preserves the distance between label vectors. Guo et al. (2019) point out that low-dimensional embedding-based models could suffer from significant overfitting. Their theoretical insights inspire a novel regularization technique to alleviate overfitting in such models. WLSTS (Evron et al., 2018) is an extreme multiclass classification framework based on *error correcting output coding*, which embeds labels with codes induced by graphs. Hsu et al. (2009) use column vectors from a matrix with the *restricted isometry property* (RIP) to represent labels. Their analysis is primarily tailored to multilabel classification. They deduce bounds for the conditional $\ell_2$-error, which measures the squared 2-norm difference between the prediction and the label vector — a metric that is not a standard measure of classification error. In contrast, our work analyzes the standard classification error.

**Label Hierarchy.** Numerous methods such as Parabel (Prabhu et al., 2018), Bonsai (Khandagale et al., 2020), AttentionXML (You et al., 2019), lightXML (Jiang et al., 2021), XR-Transformer (Zhang et al., 2021), X-Transformer (Wei et al., 2019), XR-Linear (Yu et al., 2022), and ELIAS (Gupta et al., 2022) partition the label spaces into clusters. This is typically achieved by performing $k$-means clustering on the feature space. The training process involves training a cluster-level model to assign a cluster to a feature vector, followed by training a label-level model to assign labels within the cluster.

**One-vs-all methods.** One-vs-all (OVA) algorithms address extreme classification problems with $C$ labels by modeling them as $C$ independent binary classification problems. For each label, a classifier is trained to predict its presence. DiSMEC (Babbar and Schölkopf, 2017) introduces a large-scale distributed framework to train linear OVA models, albeit at an expensive computational cost. ProXML (Babbar and Schölkopf, 2019)

---

**Meta-Algorithm 1** Label Embedding.

---

1: **Input**: dataset $\mathcal{D} = \{(x_i, y_i)\}_{i=1}^N$, embedding matrix $G$, multi-output regression algorithm $\mathcal{A}$, the decoder function $\beta^G$ from the embedding space to the label space.
2: Form the regression dataset $\mathcal{D}_r = \{(x_i, g_{y_i})\}_{i=1}^N$.
3: Train a regression model $f$ with $\mathcal{A}$ on $\mathcal{D}_r$.
4: **Return:** $\beta^G \circ f$.

---

mitigates the impact of data scarcity with adversarial perturbations. SLICE (Jain et al., 2019) accelerates negative sampling based on a generative model approximation. PD-Sparse (Yen et al., 2016) and PPD-Sparse (Yen et al., 2017a) propose optimization algorithms to exploit a sparsity assumption on labels and feature vectors.

**Other methods.** Beyond the above categories, DeepXML (Dahiya et al., 2021) uses a negative sampling procedure that shortlists $O(\log C)$ relevant labels during training and prediction. VM (Choromanska and Langford, 2015) constructs trees with $\mathcal{O}(\log C)$ depth that have leaves with low label entropy. Based on the standard random forest training algorithm, FastXML (Prabhu and Varma, 2014) proposes to directly optimize the Discounted Cumulative Gain to reduce the training cost. AnnexML (Tagami, 2017) constructs a $k$-nearest neighbor graph of the label vectors and attempts to reproduce the graph structure in a lower-dimension feature space.

## 2.2 Excess risk bounds

Our theoretical contributions are expressed as excess risk bounds, which quantify how the excess risk associated to a surrogate loss relates to the excess risk for the 0-1 loss. Excess risk bounds for classification were developed by Zhang (2004); Bartlett et al. (2006); Steinwart (2007) and subsequently developed and extended by several authors.

Ramaswamy and Agarwal (2012) shows that one needs to minimize a convex surrogate loss defined on at least a $C - 1$ dimension space to achieve consistency for the standard $C$-class classification problem, *i.e.*, any convex surrogate loss function operating on a dimension less than $C - 1$ inevitably suffers an irreducible error. Complementing this, Ávila Pires et al. (2013) validates the consistency of *simplex encoding* (Mroueh et al., 2012), a variant of label embedding in a $C - 1$ dimensional space, and introduces an excess risk bound. Previous excess risk bounds have been developed for consistent loss functions. In contrast, drawing from (Steinwart, 2007), we establish a novel excess risk bound for the label embedding framework, which admits an irreducible error and is inherently inconsistent. This error diminishes as the coherence of the embedding matrix decreases and ultimately vanishes under Massart's noise condition (Massart and Nédélec, 2006), leading to an excess risk bound of the conventional form.

From a different perspective, Ramaswamy et al. (2018) put forth a novel surrogate loss function for multiclass classification with an abstain option. This abstain option enables the classifier to opt-out from making predictions at a certain cost. Remarkably, their proposed methods not only demonstrate consistency but also effectively reduce the multiclass problems to $\lceil \log C \rceil$ binary classification problems by encoding the classes with their binary representations. In particular, the region in $\mathcal{X}$ that causes the irreducible error in our excess risk bound is abstained from in Ramaswamy et al. (2018) to achieve lossless dimension reduction in the abstention setting.

# 3 Label embedding by low-coherence matrices

Let $\mathcal{X}$ denote the feature space and $\mathcal{Y} = \{1, \ldots, C\}$ denote the label space where $C \in \mathbb{N}$. Let $(X, Y)$ be random variables in $\mathcal{X} \times \mathcal{Y}$, and let $P$ be the probability measure that governs $(X, Y)$. We use $P_{\mathcal{X}}$ to denote the marginal distribution of $P$ on $\mathcal{X}$.

We first introduce the definitions of matrix coherence in Section 3.1, followed by the notations and problem statement in Section 3.2, and then present the associated algorithm in Section 3.3. The excess risk bound and

Table 1: Frequently used symbols in Section 3

| Symbol | Description | Symbol | Description |
|---|---|---|---|
| $G$ | Embedding matrix | $\mathcal{X}$ | Feature space |
| $\mathcal{Y}$ | Label space | $P$ | Probability measure on $\mathcal{X} \times \mathcal{Y}$ |
| $P_{\mathcal{X}}$ | Marginal distribution of $P$ on $\mathcal{X}$ | $L_{01}$ | 0-1 loss function |
| $\ell^G$ | Squared loss with embedding $G$ | $\mathcal{R}_{\mathcal{L},P}$ | Risk of $\mathcal{L}$ under $P$ |
| $\mathcal{R}^*_{\mathcal{L},P}$ | Bayes risk for $\mathcal{L}$ under $P$ | $\eta(x)$ | Class posterior probabilities |
| $d(\cdot)$ | Difference between top two posteriors | $\lambda^G$ | Coherence of $G$ |
| $\beta^G$ | Decoder from embedding to label | $L^G$ | $L^G(p,y) = L_{01}(\beta^G(p), y)$ |

its interpretation are presented in Section 3.4. Finally, we introduce the condition for lossless label embedding in Section 3.5. Frequently used notations appear in Table 1.

## 3.1 Matrix coherence

Our theory relies on the notion of the coherence of a matrix $A \in \mathbb{C}^{n \times C}$, which is the maximum magnitude of the dot products between distinct columns.

**Definition 1.** Let $\{a_j\}_{j=1}^C$ be the columns of the matrix $A \in \mathbb{C}^{n \times C}$, where $\|a_j\|_2 = 1$ for all $j$. The *coherence* of $A$ is $\lambda = \max_{1 \leq i \neq j \leq C} |\langle a_i, a_j \rangle|$.

If $n \geq C$, $\lambda$ is 0 when the columns of $A$ are orthonormal. When $n < C$, however, the coherence must be positive. Indeed, Welch (1974) showed that for $A \in \mathbb{C}^{n \times C}$ and $n \leq C$, $\lambda \geq \sqrt{\frac{C-n}{n(C-1)}}$.

There are a number of known constructions of low-coherence matrices when $n < C$. A primary class of examples is the random matrices with columns of unit norms that satisfy the Johnson-Lindenstrauss property (Johnson and Lindenstraus, 1984). For example, a Rademacher matrix has entries that are sampled *i.i.d.* from a uniform distribution on $\{\frac{1}{\sqrt{n}}, -\frac{1}{\sqrt{n}}\}$. With high probability, a Rademacher random matrix of shape $n \times C$ achieves a coherence $\lambda \leq \sqrt{\frac{c_0 \log C}{n}}$ for some constant $c_0$ (Achlioptas, 2001). While random matrices can be easily obtained and have a low coherence in general, they require explicit storage (Nelson and Temlyakov, 2011) and can be outperformed in practical problems by some carefully crafted deterministic matrices (Naidu et al., 2016; Liu and Jia, 2020). Numerous deterministic constructions of low-coherence matrices have been proposed (Nelson and Temlyakov, 2011; Yu, 2011; Li et al., 2012; Xu, 2011; Naidu et al., 2016). In particular, Nelson and Temlyakov (2011) propose a deterministic construction that can achieve $\lambda \approx C^{-\frac{1}{4}}$ with $n \approx \sqrt{C}$, which avoids explicit storage of the matrix and can achieve a lower coherence in practice. There are also algorithms that directly optimize matrices for a smaller coherence (Wei et al., 2020; Abolghasemi et al., 2010; Obermeier and Martinez-Lorenzo, 2017; Li et al., 2013; Lu et al., 2018).

## 3.2 Problem statement

To define the standard classification setting, denote the 0-1 loss $L_{01} : \mathcal{Y} \times \mathcal{Y} \to \mathbb{R}$ by $L_{01}(\hat{y}, y) = \mathbb{1}_{y \neq \hat{y}}$, where $\mathbb{1}$ is the indicator function. The *risk* of a classifier $h$ is $\mathbb{E}[L_{01}(h(X), Y)]$, and the goal of classification is to learn a classifier from training data whose risk is as close as possible to the *Bayes risk* $\min_{h \in \mathcal{H}} \mathbb{E}[L_{01}(h(X), Y)]$, where $\mathcal{H} = \{\text{measurable } h : \mathcal{X} \to \mathcal{Y}\}$.

We now describe an approach to classification based on label embedding, which represents labels as vectors in $n < C$ dimensional complex space $\mathbb{C}^n$. In particular, let $G$ be an $n \times C$ matrix with unit norm columns, called the *embedding matrix*, having coherence $\lambda^G < 1$. The columns of $G$ are denoted by $g_1, g_2, \ldots, g_C$, and the column $g_i$ is used to embed the $i$-th label.

Given an embedding matrix $G$, the original $C$-class classification problem may be reduced to a multi-output regression problem, where the classification instance $(x, y)$ translates to the regression instance $(x, g_y)$. Given training data $\{(x_i, y_i)\}$ for classification, we create training data $\{(x_i, g_{y_i})\}$ for regression, and apply any algorithm for multi-output regression to learn a regression function $f : \mathcal{X} \to \mathbb{C}^n$.

At test time, given a test point $x$, a label $y$ is obtained by taking the nearest neighbor to $f(x)$ among the columns of $G$. In particular, define the decoding function $\beta^G : \mathbb{C}^n \to \mathcal{Y}$, $\beta^G(p) = \min\{\arg\min_{i \in \mathcal{Y}} \|p - g_i\|_2\}$, where $p$ represents the output of a regression model. (Since the arg min is potentially set-valued, the min breaks ties in favor of the label with the smallest index.) Then the label assigned to $x$ is $\beta^G(f(x))$.

Thus, label embedding gives rise to classifiers of the form $\beta^G \circ f$, where $f \in \mathcal{F} = \{$all measurable $f : \mathcal{X} \to \mathbb{C}^n\}$. Fortunately, according to the following result, no expressiveness is lost by considering classifiers of this form.

**Proposition 2.** *Recall the decoding function* $\beta^G(p) = \min\{\arg\min_{i \in \mathcal{Y}} \|p - g_i\|_2\}$, *the regression models* $\mathcal{F} = \{$all measurable $f : \mathcal{X} \to \mathbb{C}^n\}$, *and the classification models* $\mathcal{H} = \{$all measurable $h : \mathcal{X} \to \mathcal{Y}\}$. *Then* $\beta^G \circ \mathcal{F} = \mathcal{H}$.

While the composition $\beta^G \circ \mathcal{F}$ preserves expressiveness, choosing a low-dimensional embedding in conjunction with a convex surrogate can introduce irreducible error (Ramaswamy and Agarwal, 2012).

This and other results are proved in an appendix. It follows that $\min_{f \in \mathcal{F}} \mathbb{E}_P\left[L_{01}(\beta^G(f(X)), Y)\right]$ is the Bayes risk for classification as defined earlier. This allows us to focus our attention on learning $f : \mathcal{X} \to \mathbb{C}^n$. Toward that end, we now formalize notions of loss and risk for the task of learning a multi-output function $f$ for multiclass classification via label embedding.

**Definition 3.** A loss function for label embedding is a function $\mathcal{L} : \mathbb{C}^n \times \mathcal{Y} \to \mathbb{R}$. Given such a loss function, define the $\mathcal{L}$-risk of $f$ with distribution $P$ to be $\mathcal{R}_{\mathcal{L},P} : \mathcal{F} \to \mathbb{R}$, $\mathcal{R}_{\mathcal{L},P}(f) := \mathbb{E}_P[\mathcal{L}(f(X), Y)]$ and the $\mathcal{L}$-Bayes risk to be $\mathcal{R}^*_{\mathcal{L},P} := \inf_{f \in \mathcal{F}} \mathcal{R}_{\mathcal{L},P}(f)$.

Using this notation, the *target* loss for learning with a fixed embedding matrix $G$ is the loss function $L^G : \mathbb{C}^n \times \mathcal{Y}$ defined by $L^G(p, y) := L_{01}(\beta^G(p), y)$. By Prop. 2, the $L^G$-risk of $f$ is the usual classification risk of the associated classifier $h = \beta^G \circ f$, and the $L^G$-Bayes risk is the Bayes risk of the original classification problem. The goal is to find a pair $(f, G)$ that minimizes $\mathcal{R}_{L^G,P}(f)$, or in other words, a pair $(f, G)$ that makes the *excess target risk* $\mathcal{R}_{L^G,P} - \mathcal{R}^*_{L^G,P}$ as small as possible.

While the target loss $L^G$ defines the learning goal, it is not practical as a training objective because of its discrete nature. Therefore, for learning purposes, Hsu et al. (2009); Akata et al. (2013); Yu et al. (2014) suggest a *surrogate* loss, namely, the squared distance between $f(x)$ and $g_y$. More precisely, for a given embedding matrix $G$, define $\ell^G : \mathbb{C}^n \times \mathcal{Y} \to \mathbb{R}$ as $\ell^G(p, y) := \frac{1}{2}\|p - g_y\|_2^2$. This surrogate allows us to learn $f$ by applying existing multi-output regression algorithms as we describe next. Thus, the label embedding learning problem is to learn $f$ with small surrogate excess risk. Our subsequent analysis will connect $\mathcal{R}_{L^G,P} - \mathcal{R}^*_{L^G,P}$ to $\mathcal{R}_{\ell^G,P} - \mathcal{R}^*_{\ell^G,P}$.

### 3.3 Learning algorithms

Learning algorithms for classification via label embedding, as described thus far, can be summarized by a conceptually simple meta-algorithm, depicted in Meta-Algorithm 1. This meta-algorithm should not be considered novel as its essential ingredients have been previously introduced Akata et al. (2013); Rodríguez et al. (2018), and several existing algorithms can be seen as instances (Hsu et al., 2009; Yu et al., 2014; Bhatia et al., 2015; Evron et al., 2018; Akata et al., 2013).

The meta-algorithm takes as input a training dataset $\{(x_i, y_i)\}_{i=1}^N$, an embedding matrix $G = [g_1, g_2, \ldots, g_C]$, and an algorithm $\mathcal{A}$ for multi-output regression. It forms the multi-output regression dataset $\{(x_i, g_{y_i})\}_{i=1}^N$, and applies $\mathcal{A}$ to produce a function $f$. The output is the classifier $\beta^G \circ f$.

For example, the regression algorithm can be specified by selecting a model class $\mathcal{F}_0$ and a surrogate loss $\ell^G$, and learning $f$ by empirical risk minimization:

$$\min_{f \in \mathcal{F}_0} \frac{1}{N} \sum_{i=1}^N \ell^G(f(x_i), y_i).$$

In our experiments we select $\mathcal{F}_0$ to be a neural network with $n$ nodes in the output layer, and $\ell^G$ to be the squared error loss mentioned previously, the same surrogate analyzed in the next section.

As a remark, we have treated $G$ as a fixed input to the meta-algorithm, but it can also be trained jointly with $f$. Our analysis is independent of model training, and thus applies to this case as well.

In the next section we present theory that supports selecting $G$ with low coherence, which has not previously been examined in the label embedding literature.

### 3.4 Excess risk bound

We present an excess risk bound, which relates the excess surrogate risk to the excess target risk. This bound justifies the use of the squared error surrogate, and also reveals a trade-off between the reduction in dimensionality (as reflected by $\lambda^G$) and the potential penalty in accuracy.

To state the bound, define the class posterior $\eta(x) = (\eta_1(x), \ldots, \eta_C(x))$ where $\eta_i(x) = P_{Y|X=x}(i)$. Define $d(x) = \max_i \eta_i(x) - \max_{i \notin \arg\max_j \eta_j(x)} \eta_i(x)$, which is a measure of "margin" at $x$. We discuss this quantity further after the main result, which we now state.

**Theorem 4.** *Consider an embedding matrix $G$ with unit norm columns $g_1, g_2, \ldots, g_C$ and coherence $\lambda^G = \max_{i \neq j} |\langle g_i, g_j \rangle|$. Recall $\mathcal{R}_{L^G,P}$ and $\mathcal{R}_{\ell^G,P}$ represent risks as defined in Definition 3, with $\mathcal{R}^*_{L^G,P}$ and $\mathcal{R}^*_{\ell^G,P}$ being the corresponding Bayes risks. Then for all $f \in \mathcal{F}$,*

$$
\mathcal{R}_{L^G,P}(f) - \mathcal{R}^*_{L^G,P} \leq \inf_{r > \frac{2\lambda^G}{1+\lambda^G}} \left\{ \frac{2\lambda^G}{1+\lambda^G} P_{\mathcal{X}}(d(X) < r) \right.
$$
$$
+ \sqrt{\frac{4 - 2\lambda^G}{(1+\lambda^G)^2} P_{\mathcal{X}}(d(X) < r)(\mathcal{R}_{\ell^G,P}(f) - \mathcal{R}^*_{\ell^G,P})}
$$
$$
\left. + \frac{4 - 2\lambda^G}{(r(1+\lambda^G) - 2\lambda^G)^2} \left( \mathcal{R}_{\ell^G,P}(f) - \mathcal{R}^*_{\ell^G,P} \right) \right\}
$$

As mentioned earlier, the goal of learning is to minimize the excess target risk $\mathcal{R}_{L^G,P}(f) - \mathcal{R}^*_{L^G,P}$. The theorem shows that this goal can be achieved up to the first (irreducible) term by minimizing the excess surrogate risk $\mathcal{R}_{\ell^G,P}(f) - \mathcal{R}^*_{\ell^G,P}$. The excess surrogate risk can be driven to zero by any consistent algorithm for multi-output regression with squared error loss. We provide a formal definition of consistency and illustrate it with examples of both consistent and inconsistent loss functions in the appendix.

The quantity $d(x)$ can be viewed as a measure of noise (inherent in the joint distribution of $(X, Y)$) at a point $x$. While $\max_i \eta_i(x)$ represents the probability of the most likely label occurring, $\max_{i \notin \arg\max_j \eta_j(x)} \eta_i(x)$ represents the probability of the second most likely label occurring. A large $d(x)$ implies that $\arg\max_i \eta_i(x)$ is, with high confidence, the correct prediction at $x$. In contrast, if $d(x)$ is small, our confidence in predicting $\arg\max_i \eta_i(x)$ is reduced, as the second most likely label has a similar probability of being correct.

As pointed out by Ramaswamy and Agarwal (2012), any convex surrogate loss function operating on a dimension less than $C - 1$ inevitably suffers an irreducible error. In the present setting, this irreducible error is manifested in the first term, which depends on the coherence $\lambda^G$ of the embedding matrix. Given a classification problem with $C$ classes, a larger embedding dimension $n$ will lead to a smaller coherence $\lambda^G$, making $d(x) > \frac{2\lambda^G}{1+\lambda^G}$ on a larger region in $\mathcal{X}$ at the cost of increasing computational complexity. On the other hand, by choosing a smaller $n$, $d(x) < \frac{2\lambda^G}{1+\lambda^G}$ on a larger region in $\mathcal{X}$, increasing the first term in Theorem 4. This interpretation highlights the balance between the benefits of dimensionality reduction and the potential impact on prediction accuracy, as a function of the coherence of the embedding matrix, $\lambda^G$, and the noisiness measure, $d(x)$.

### 3.5 Improvement under low noise

While Theorem 4 holds universally (for all distributions $P$), by considering a specific subset of distributions, we can derive a more conventional form of the excess risk bound. As a direct consequence of Theorem 4, under the multiclass extension of the Massart noise condition (Massart and Nédélec, 2006), which requires $d(X) > c$ with probability 1 for some $c$, the first and second terms in Theorem 4 vanish. In this case, we

recover a conventional excess risk bound, where $\mathcal{R}_{L^G,P}(f) - \mathcal{R}^*_{L^G,P}$ tends to 0 with $\mathcal{R}_{\ell^G,P}(f) - \mathcal{R}^*_{\ell^G,P}$. We now formalize this.

**Definition 5** (Multiclass Massart Noise Condition). *The distribution $P$ on $\mathcal{X} \times \mathcal{Y}$ is said to satisfy the Multiclass Massart Noise Condition if and only if $\exists c > 0$ such that $P_{\mathcal{X}}(d(X) \geq c) = 1$.*

**Corollary 6.** *Consider the same setup as in Theorem 4 and assume $P$ satisfies the Multiclass Massart Noise Condition. If $\lambda^G \in \left(0, \frac{\operatorname{ess\,inf} d}{2 - \operatorname{ess\,inf} d}\right)$, then for all $f \in \mathcal{F}$*

$$\mathcal{R}_{L^G,P}(f) - \mathcal{R}^*_{L^G,P} \leq \frac{4 - 2\lambda^G}{((1 + \lambda^G)\operatorname{ess\,inf} d - 2\lambda^G)^2}\left(\mathcal{R}_{\ell^G,P}(f) - \mathcal{R}^*_{\ell^G,P}\right),$$

*where $\operatorname{ess\,inf} d$ is the essential infimum of $d$, i.e., $\operatorname{ess\,inf} d = \sup\{a \in \mathbb{R} : P_{\mathcal{X}}(d(X) < a) = 0\}$.*

For the special case where all labels are deterministic, we have $\operatorname{ess\,inf} d(x) = 1$ for all $x$, leading to the simplified bound

$$\mathcal{R}_{L^G,P}(f) - \mathcal{R}^*_{L^G,P} \leq \frac{4 - 2\lambda^G}{(1 - \lambda^G)^2}(\mathcal{R}_{\ell^G,P}(f) - \mathcal{R}^*_{\ell^G,P}).$$

This observation implies that for deterministic labels, any embedding matrix with coherence less than 1 ensures consistency. Furthermore, a smaller coherence means faster convergence.

## 4 Experiments

Table 2: Summary of the datasets used in the experiments. $N_{\text{train}}$ is the number of training data points, $N_{\text{test}}$ the number of test data points, $D$ the number of features, and $C$ the number of classes.

| Dataset | $N_{\text{train}}$ | $N_{\text{test}}$ | $D$ | $C$ |
|---------|--------|--------|--------|--------|
| LSHTC1 | 83805 | 5000 | 328282 | 12046 |
| DMOZ | 335068 | 38340 | 561127 | 11879 |
| ODP | 975936 | 493014 | 493014 | 103361 |

In this section, we present an experimental evaluation of our proposed method, LOCOLE (LOw COherence Label Embedding), for extreme multiclass classification. LOCOLE is an instance of Meta-Algorithm 1, as we explain below.

### 4.1 Experiment setup

We conduct experiments on three large-scale datasets, DMOZ (Partalas et al., 2015), LSHTC1 (Partalas et al., 2015), and ODP (Bennett and Nguyen, 2009), which are extensively used for benchmarking extreme classification algorithms. The details of these datasets are provided in Table 2, with DMOZ and LSHTC1 available from (Yen et al., 2016)[1], and ODP from (Medini et al., 2019).

We apply LOCOLE where the multi-output regression algorithm is to train a multilayer perceptron with $n$ output layer nodes using the surrogate loss $\ell^G$. This is implemented using PyTorch, with a 2-layer fully connected neural network used for the LSHTC1 and DMOZ datasets and a 4-layer fully connected neural network for the ODP dataset. The hyperparameters are tuned on a held-out dataset.

We experiment with the following types of embedding matrices:

- Rademacher: entries sampled *i.i.d.* from a uniform distribution on $\{\frac{1}{\sqrt{n}}, -\frac{1}{\sqrt{n}}\}$.

- Gaussian: entries sampled *i.i.d.* from $\mathcal{N}(0, \frac{1}{n})$. Columns are normalized to have unit norm.

- $\mathbb{C}$-Gaussian: the real and imaginary parts of each entry are sampled *i.i.d.* from $\mathcal{N}(0, \frac{1}{2n})$. Each column is normalized to have unit norm.

---

[1] https://people.csail.mit.edu/xrhuang/PDSparse/index.html

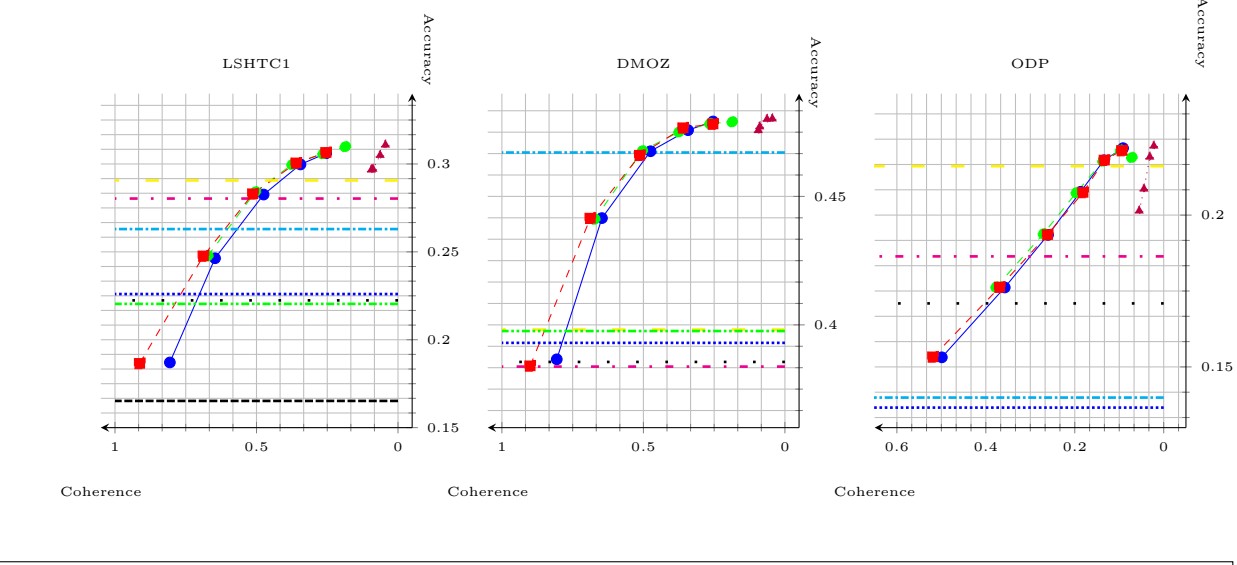

Figure 1: These plots reveal an inverse correlation between embedding matrix coherence and classification accuracy across different datasets, with coherence on the horizontal axis and accuracy on the vertical. Non-LOCOLE methods are plotted as horizontal lines because they do not depend on the embedding dimensionality.

- Nelson (Nelson and Temlyakov, 2011): a deterministic construction of low-coherence complex matrices in which $n$ must be prime. If $r$ is an integer, $n > r$ is a prime number, and $n^r \geq C$, then the coherence of the constructed matrix is at most $\frac{r-1}{\sqrt{n}}$. We choose $r = 2$ in experiments.

We compare LOCOLE against the following state-of-the-art methods:

- PD-Sparse (Yen et al., 2016): an efficient solver designed to exploit the sparsity in extreme classification.

- PPD-Sparse (Yen et al., 2017a): a multi-process extension of PD-Sparse (Yen et al., 2016).

- Parabel (Prabhu et al., 2018): a tree-based method which builds a label-hierarchy.

- AnnexML (Tagami, 2017): a method which constructs a $k$-nearest neighbor graph of the label vectors and attempts to reproduce the graph structure from a lower-dimension feature space.

- WLSTS (Evron et al., 2018): a method based on *error correcting output coding* which embeds labels by codes induced by graphs.

- MLP (CE) Standard multilayer perceptron classifier with cross-entropy loss.

- MLP (SE): Standard multilayer perceptron classifier with squared error loss.

For these methods, we use the hyperparameters suggested by their papers or accompanying code.

While there are numerous label embedding algorithms (Hsu et al., 2009; Yu et al., 2014; Bhatia et al., 2015; Evron et al., 2018; Akata et al., 2013) which could be seen as specific subsets or instances of Meta-Algorithm 1, our comparisons will not include all of them. The approach in Akata et al. (2013) is not tailored for extreme classification and requires auxiliary information to construct the embedding matrices. On the other hand, methods like Yu et al. (2014); Bhatia et al. (2015); Hsu et al. (2009) have been outperformed significantly by the current state-of-the-art algorithms as shown in prior work (Yen et al., 2016; Bengio et al., 2010).

Table 3: Accuracy on Various Datasets. RRM denotes **R**ademacher **R**andom **M**atrices, GRM stands for **G**aussian **R**andom **M**atrices, and ℂGRM stands for Complex **G**aussian **R**andom **M**atrices. Methods without an embedding dimension ('-Dim') are non-LOCOLE methods. We stop training when the training time reaches 50 hours and indicate this in the table.

| **LSHTC1** | | **DMOZ** | | **ODP** | |
|---|---|---|---|---|---|
| Method(-Dim) | Acc.(Mean±Std) | Method(-Dim) | Acc.(Mean±Std) | Method(-Dim) | Acc.(Mean±Std) |
| GRM–32 | $18.71 \pm 0.22\%$ | GRM–32 | $38.66 \pm 0.18\%$ | GRM–256 | $17.62 \pm 0.07\%$ |
| GRM–64 | $24.64 \pm 0.23\%$ | GRM–64 | $44.16 \pm 0.01\%$ | GRM–512 | $19.35 \pm 0.04\%$ |
| GRM–128 | $28.26 \pm 0.19\%$ | GRM–128 | $46.76 \pm 0.05\%$ | GRM–1024 | $20.77 \pm 0.03\%$ |
| GRM–256 | $29.97 \pm 0.16\%$ | GRM–256 | $47.58 \pm 0.12\%$ | GRM–2048 | $21.81 \pm 0.07\%$ |
| GRM–512 | $30.61 \pm 0.22\%$ | GRM–512 | $47.92 \pm 0.07\%$ | GRM–4096 | $22.20 \pm 0.04\%$ |
| ℂGRM–32 | $24.77 \pm 0.31\%$ | ℂGRM–32 | $44.11 \pm 0.04\%$ | ℂGRM–256 | $19.39 \pm 0.07\%$ |
| ℂGRM–64 | $28.42 \pm 0.22\%$ | ℂGRM–64 | $46.78 \pm 0.07\%$ | ℂGRM–512 | $20.74 \pm 0.08\%$ |
| ℂGRM–128 | $29.97 \pm 0.21\%$ | ℂGRM–128 | $47.51 \pm 0.09\%$ | ℂGRM–1024 | $21.78 \pm 0.04\%$ |
| ℂGRM–256 | $30.58 \pm 0.13\%$ | ℂGRM–256 | $47.84 \pm 0.06\%$ | ℂGRM–2048 | $22.14 \pm 0.06\%$ |
| ℂGRM–512 | $30.98 \pm 0.14\%$ | ℂGRM–512 | $47.90 \pm 0.08\%$ | ℂGRM–4096 | $21.90 \pm 0.07\%$ |
| RRM-32 | $18.66 \pm 0.20\%$ | RRM-32 | $38.41 \pm 0.17\%$ | RRM-256 | $17.63 \pm 0.04\%$ |
| RRM-64 | $24.77 \pm 0.17\%$ | RRM-64 | $44.15 \pm 0.05\%$ | RRM-512 | $19.36 \pm 0.04\%$ |
| RRM-128 | $28.30 \pm 0.27\%$ | RRM-128 | $46.60 \pm 0.04\%$ | RRM-1024 | $20.75 \pm 0.05\%$ |
| RRM-256 | $30.06 \pm 0.21\%$ | RRM-256 | $47.67 \pm 0.09\%$ | RRM-2048 | $21.81 \pm 0.06\%$ |
| RRM-512 | $30.66 \pm 0.22\%$ | RRM-512 | $47.82 \pm 0.06\%$ | RRM-4096 | $22.13 \pm 0.08\%$ |
| Nelson-113 | $29.66 \pm 0.12\%$ | Nelson-113 | $47.57 \pm 0.06\%$ | Nelson-331 | $20.14 \pm 0.06\%$ |
| Nelson-127 | $29.71 \pm 0.13\%$ | Nelson-127 | $47.71 \pm 0.04\%$ | Nelson-509 | $20.86 \pm 0.09\%$ |
| Nelson-251 | $30.50 \pm 0.16\%$ | Nelson-251 | $48.01 \pm 0.04\%$ | Nelson-1021 | $21.91 \pm 0.06\%$ |
| Nelson-509 | $31.08 \pm 0.15\%$ | Nelson-509 | $48.02 \pm 0.06\%$ | Nelson-2039 | $22.30 \pm 0.06\%$ |
| MLP (CE) | $26.30 \pm 0.36\%$ | MLP (CE) | $46.71 \pm 0.06\%$ | MLP (CE) | $13.99 \pm 0.11\%$ |
| MLP (SE) | $28.03 \pm 0.17\%$ | MLP (SE) | $38.38 \pm 0.14\%$ | MLP (SE) | $18.64 \pm 0.02\%$ |
| Annex ML | $29.06 \pm 0.35\%$ | Annex ML | $39.82 \pm 0.14\%$ | Annex ML | $21.61 \pm 0.04\%$ |
| Parabel | $22.24 \pm 0.00\%$ | Parabel | $38.56 \pm 0.00\%$ | Parabel | $17.09 \pm 0.00\%$ |
| WLSTS | $16.52 \pm 1.43\%$ | WLSTS | $13.60 \pm 1.49\%$ | WLSTS | Train > 50 hrs |
| PDSparse | $22.04 \pm 0.06\%$ | PDSparse | $39.76 \pm 0.03\%$ | PDSparse | Train > 50 hrs |
| PPDSparse | $22.60 \pm 0.11\%$ | PPDSparse | $39.30 \pm 0.07\%$ | PPDSparse | $13.66 \pm 0.05\%$ |

Table 4: Accuracy and training time across methods. See Section 4.3 for details.

| Dataset | Metric | Single Node | | Distributed | |
|---|---|---|---|---|---|
| | | PD-Sparse | LOCOLE | PPD-Sparse | LOCOLE |
| *LSHTC1* | Accuracy | 22.04% | 23.42% | 22.60% | 23.38% |
| | Training Time | 230s | 55s | 135s | 14s |
| *DMOZ* | Accuracy | 39.76% | 41.09% | 39.30% | 40.57% |
| | Training Time | 829s | 254s | 656s | 68s |
| *ODP* | Accuracy | N/A | 15.11% | 13.66% | 15.06% |
| | Training Time | > 50 hrs | 2045s | 668s | 350s |

While our theoretical framework is adaptable to any form of embedding, whether trained, fixed, or derived from auxiliary information, we focus on fixed embeddings in our empirical studies. This choice stems from the absence of a standardized or widely accepted methodology for jointly training the embedding function and the classifier. Although heuristic approaches exist, we are concerned that they may introduce confounding

factors, complicating empirical interpretation. By centering on fixed embeddings, we ensure a controlled evaluation, minimizing such confounding factors and emphasizing the role of coherence of the embeddings.

All neural network training is performed on a single NVIDIA A40 GPU with 48GB RAM. We explore different embedding dimensions and provide figures showing the relationship between the coherence of $G$ and the accuracy. Full experimental details are presented in section B in the appendix.

### 4.2 Experimental results

The experimental results in Table 3 highlight the superior performance of our proposed method across various datasets. In Table 3, the column Method (-Dim) denotes the method or embedding type along with its dimension, while the Acc. (Mean $\pm$ Std) column presents the mean and standard deviation of accuracy over 5 randomized repetitions. Methods without an embedding dimension (-Dim) are non-LOCOLE methods, which are unaffected by the embedding dimension. We highlight the best-performing method for each dataset. For each embedding method, there is a clear trend of increasing accuracy with larger embedding dimension. Ultimately, every embedding type outperforms all non-LOCOLE methods as the embedding dimension increases, with the Nelson embedding at dimension 509 achieving the highest accuracies across all datasets. We stop the training when the training time reaches 50 hours and indicate this in the table.

We plot the accuracies as the coherence of the embedding matrix decreases in Figure 1 for the LSHTC1, DMOZ, and ODP datasets. Alongside, we include several baselines for comparison. Figure 1 demonstrates a negative correlation between the coherence of the embedding matrix and the accuracy, confirming our theoretical analysis. We include plots of coherence vs precision and recall in Section B of the appendix, along with low-dimensional projections of the embedding vectors.

### 4.3 Computational advantage

PD-Sparse (Yen et al., 2016) and PPD-Sparse (Yen et al., 2017b) are among the most computationally efficient methods for training for extreme classification, to the best of our knowledge. PD-Sparse and PPD-Sparse both efficiently fit a linear model for classification with a multiclass hinge loss and elastic net regularization, which is regularization with both $\ell_1$ and $\ell_2$ penalties. For comparison, we apply LOCOLE using the Rademacher embedding to elastic net-regularized (Zou and Hastie, 2005) linear regression with $\ell^G$ loss. To compare the computational efficiency, we set the embedding dimension to $n = 360$. In our distributed implementation (multi-output linear regression can be trivially parallelized by solving multiple scalar-output linear regressions), each node independently solves a subset of elastic net linear regressions with scalar output, effectively spreading out the computation. In Table 4, we compare LOCOLE with Rademacher embedding with PD-sparse and PPD-sparse. LOCOLE clearly outperforms PD-sparse and PPD-sparse in both runtime and accuracy. We train the PD-Sparse method and Single-Node LOCOLE on Intel Xeon Gold 6154 processors, equipped with 36 cores and 180GB of memory. The distributed LOCOLE and the PPD-Sparse method — also implemented in a distributed fashion — are trained across 10 CPU nodes, harnessing 360 cores and 1.8TB of memory in total.

## 5 Conclusion and future work

We provide a theoretical analysis for label embedding methods in the context of extreme multiclass classification. Our analysis confers a deeper understanding of the tradeoffs between dimensionality reduction and accuracy. We derive an excess risk bound that quantifies this tradeoff in terms of the coherence of the embedding matrix, and show that the statistical penalty for label embedding vanishes under the multiclass Massart condition. Through extensive experiments, we demonstrated that label embedding with low-coherence matrices outperforms existing techniques in both accuracy and runtime.

While our analysis focuses on excess risk, the reduction of classification to regression means that existing generalization error bounds (for multi-output regression with squared error loss) can be applied to analyze the generalization error in our context. For example, Reeve and Kabán (2020) show that the generalization

error grows with the dimension of the output space. This suggests that smaller embedding dimension leads to tighter control of the generalization error.

Building on out theoretical framework, future work may consider extensions to multilabel classification, online learning, zero-shot learning, and learning with rejection.

## 6 Limitations

Lossless label embedding relies on Massart's noise condition. Although Massart's condition is a well-recognized assumption in learning theory, it is important to note that it remains a theoretical construct that cannot be directly verified in most practical scenarios. This assumption facilitates theoretical analysis but may not always reflect real-world data distributions.

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

# A  Proofs

In this section, we present the proofs of the results in the main paper.

## A.1  Proof of proposition 2

*Proof.* This stems from the following facts: (i) for any given $f \in \mathcal{F}$, the function $\beta^G \circ f$ is also measurable, i.e., $\beta^G \circ \mathcal{F} \subset \mathcal{H}$, (ii) for all $h \in \mathcal{H}$, the function $f(x) = g_{h(x)}$ ensures that $\beta^G \circ f = h$, and (iii) $f(x) = g_{h(x)}$ is measurable as it only attains finite number of values. So (ii) and (iii) imply $\mathcal{H} \subset \beta^G \circ \mathcal{F}$.  □

## A.2  Proof of the main theorem

Recall that $\beta^G(p) = \min\{\arg\min_{i \in \mathcal{Y}} \|p - g_i\|_2\}$.

**Lemma 7.** $\forall p \in \mathbb{C}^n, \forall x \in \mathcal{X}, C_{1,x}(p) = \max_i \eta_i(x) - \eta_{\beta^G(p)}(x)$. .

*Proof.* Recall that $L^G(p, y) = L_{01}(\beta^G(p), y) = \mathbb{1}_{\{\min\{\arg\min_{i \in \mathcal{Y}} \|p - g_i\|_2\} \neq y\}}$. The result follows from the observation that $C_{L^G,x}(p) = 1 - \eta_{\beta^G(p)}(x)$ and $C^*_{L^G,x} = 1 - \max_i \eta_i(x)$.  □

**Lemma 8.** $C_{2,x}$ is strictly convex and minimized at $p^*_x = G\eta(x) = \sum_{i=1}^C \eta_i(x)g_i$.

*Proof.*

$$C_{\ell^G,x}(p) = \mathbb{E}_{y \sim P_{Y|X=x}} \ell_2(p, g_y)$$

$$= \frac{1}{2}\sum_{i=1}^C \eta_i(x)\|p - g_i\|_2^2$$

$$= \frac{1}{2}\sum_{i=1}^C \eta_i(x)\langle p - g_i, p - g_i \rangle$$

$$= \frac{1}{2}\langle p, p\rangle - \mathfrak{Re}\left\langle \sum_{i=1}^C \eta_i(x)g_i, p \right\rangle + \frac{1}{2}\sum_{i=1}^C \eta_i(x)\|g_i\|_2^2$$

$$= \frac{1}{2}\left\| p - \sum_{i=1}^C \eta_i(x)g_i \right\|_2^2 + \frac{1}{2}\sum_{i=1}^C \eta_i(x)\|g_i\|_2^2 - \frac{1}{2}\left\| \sum_{i=1}^C \eta_i(x)g_i \right\|_2^2$$

So $C_{2,x}(p)$ is strictly convex and minimized at $p^*_x = \sum_{i=1}^C \eta_i(x)g_i = G\eta(x)$.  □

We'll use $p^*_x$ to denote $G\eta(x)$ henceforward.

**Lemma 9.** *Let $V$ be a normed space with norm $\|\cdot\|_V$. Let $u : V \to \mathbb{R}$ be a strictly convex function. Let $u$ be minimized at $x^*$ with $u(x^*) = 0$. $\forall x \in V, \forall \delta_0 > 0$, if $u(x) < \delta := \inf_{x: \|x-x^*\|_V = \delta_0} u(x)$, then $\|x - x^*\|_V < \delta_0$.*

*Proof.* We first confirm the fact that $\forall t \in (0,1)$ and $q \in V - \{0\}$, $u(x^* + tq) < u(x^* + q)$.

$$u(x^* + tq) = u((1-t)x^* + t(x^* + q))$$
$$< (1-t)u(x^*) + tu(x^* + q)$$
$$= tu(x^* + q)$$
$$< u(x^* + q).$$

Assume the opposite: For some $u \in V$, $\delta_0 > 0$, $u(x) < \delta = \inf_{x: \|x-x^*\|_V = \delta_0} u(x)$ and $\|x - x^*\|_V \geq \delta_0$. Then

$$u(x) = u(x^* + (x - x^*)) \geq u\left(x^* + \frac{\delta_0}{\|x - x^*\|_V}(x - x^*)\right) \geq \delta,$$

which results in a contradiction.  □

**Lemma 10.** *Fix $x \in \mathcal{X}$. $\forall \delta_0 > 0$ $\forall p \in \mathbb{C}^n$, $C_{2,x}(p) < \frac{1}{2}\delta_0^2 \implies \|p - p_x^*\| < \delta_0$.*

*Proof.* Applying Lemma 9 with $u = C_{2,x}$, $V = \mathbb{C}^n$, $\|\cdot\|_V = \|\cdot\|_2$, and $x^* = p_x^*$, we have $\forall p \in \mathbb{C}^n$, $C_{2,x}(p) < \inf_{s:\|s-p_x^*\|=\delta_0} C_{2,x}(s) \implies \|p - p_x^*\|_2 < \delta_0$. Furthermore,

$$
\begin{aligned}
\inf_{s:\|s-p_x^*\|=\delta_0} C_{2,x}(s) &= \inf_{s:\|s-p_x^*\|=\delta_0} \left( \frac{1}{2}\langle s, s \rangle - \mathfrak{Re}\langle p_x^*, s \rangle \right) - \left( \frac{1}{2}\langle p_x^*, p_x^* \rangle - \mathfrak{Re}\langle p_x^*, p_x^* \rangle \right) \\
&= \inf_{s:\|s-p_x^*\|=\delta_0} \frac{1}{2}\|s - p_x^*\|_2^2 \\
&= \frac{1}{2}\delta_0^2.
\end{aligned}
$$

$\square$

To facilitate our proofs, we denote $\lambda_{j,k} = \langle g_j, g_k \rangle$, $\lambda_{j,k}^{\mathfrak{Re}} = \mathfrak{Re}\lambda_{j,k}$, and $\lambda^G = \max_{j \neq k} |\lambda_{j,k}|$.

**Lemma 11.** *$\forall x \in \mathcal{X}$, $\forall j, k \in [C]$, $\forall p \in \mathbb{C}^n$,*

$$
\frac{(1 + \lambda^G)(\eta_k(x) - \eta_j(x)) - 2\lambda^G}{\sqrt{2 - 2\lambda^G}} > \|p_x^* - p\|_2 \implies \|p - g_j\|_2 > \|p - g_k\|_2.
$$

*Proof.* We first consider the general case when $p \neq p_x^*$. Write $p = p_x^* + \delta_0 v$ where $v = \frac{p - p_x^*}{\|p - p_x^*\|_2}$ and $\delta_0 = \|p - p_x^*\|_2$. Recall that $p_x^* = G\eta(x)$. Note the inequality immediately implies $\eta_k(x) > \eta_j(x)$.

$$
\frac{(1 + \lambda^G)(\eta_k(x) - \eta_j(x)) - 2\lambda^G}{\sqrt{2 - 2\lambda^G}} > \delta_0
$$

$$
\implies (1 - \lambda^G)(\eta_k(x) - \eta_j(x)) - 2\lambda^G(1 - (\eta_k(x) - \eta_j(x))) > \delta_0\sqrt{2 - 2\lambda^G}
$$

$$
\implies (1 - \lambda^G)(\eta_k(x) - \eta_j(x)) - 2\lambda^G(1 - \eta_k(x) - \eta_j(x)) > \delta_0\sqrt{2 - 2\lambda^G}
$$

$$
\implies \sqrt{1 - \lambda^G}(\eta_k(x) - \eta_j(x)) - \frac{2\lambda^G}{\sqrt{1 - \lambda^G}}(1 - \eta_k(x) - \eta_j(x)) > \sqrt{2}\delta_0
$$

$$
\implies \sqrt{1 - \lambda_{j,k}^{\mathfrak{Re}}}(\eta_k(x) - \eta_j(x)) + \frac{1}{\sqrt{1 - \lambda_{j,k}^{\mathfrak{Re}}}} \sum_{i \neq j,k} (\lambda_{i,k}^{\mathfrak{Re}} - \lambda_{i,j}^{\mathfrak{Re}})\eta_i(x) > \sqrt{2}\delta_0 \tag{1}
$$

$$
\implies (1 - \lambda_{j,k}^{\mathfrak{Re}})(\eta_k(x) - \eta_j(x)) + \sum_{i \neq j,k} (\lambda_{i,k}^{\mathfrak{Re}} - \lambda_{i,j}^{\mathfrak{Re}})\eta_i(x) > \delta_0\sqrt{2 - 2\lambda_{j,k}^{\mathfrak{Re}}}
$$

$$
\implies \mathfrak{Re}\langle p^*, g_k - g_j \rangle > \delta_0 \|g_j - g_k\|_2 \tag{2}
$$

$$
\implies \mathfrak{Re}\langle p^*, g_k - g_j \rangle > \mathfrak{Re}\langle \delta_0 v, g_j - g_k \rangle \tag{3}
$$

$$
\implies \mathfrak{Re}\langle p^* + \delta_0 v, g_k \rangle > \mathfrak{Re}\langle p^* + \delta_0 v, g_j \rangle
$$

$$
\implies \|p - g_j\|_2^2 > \|p - g_k\|_2^2 \tag{4}
$$

Inequality (1) follows the fact that $\forall i, i', |\lambda_{i,i'}^{\mathfrak{Re}}| \leq \lambda^G$. Inequality (2) follows from $p_x^* = \sum_{i=1}^C \eta_i(x)g_i$ and $\|g_j - g_k\|_2 = \sqrt{2 - 2\lambda_{j,k}^{\mathfrak{Re}}}$. Inequality (3) is implied by the Cauchy-Schwarz inequality. In the last inequality (4), we use the fact that $\|g_j\|_2 = \|g_k\|_2 = 1$.

Now let $p = p_x^*$. Let $\frac{(1+\lambda^G)(\eta_k(x) - \eta_j(x)) - 2\lambda^G}{\sqrt{2-2\lambda^G}} > 0$.

$$
\begin{aligned}
&\|p - g_j\|_2^2 - \|p - g_k\|_2^2 \\
=&2\mathfrak{Re}\langle p_x^*, g_k - g_j \rangle \\
=&2(1 - \lambda_{j,k}^{\mathfrak{Re}})(\eta_k(x) - \eta_j(x)) + 2\sum_{i \neq j,k}(\lambda_{i,k}^{\mathfrak{Re}} - \lambda_{i,j}^{\mathfrak{Re}})\eta_i(x) \\
\geq&2(1 - \lambda^G)(\eta_k(x) - \eta_j(x)) - 4\lambda^G(1 - \eta_k(x) - \eta_j(x)) \\
\geq&2(1 - \lambda^G)(\eta_k(x) - \eta_j(x)) - 4\lambda^G(1 - (\eta_k(x) - \eta_j(x))) \\
=&2(1 + \lambda^G)(\eta_k(x) - \eta_j(x)) - 4\lambda^G > 0
\end{aligned}
$$

$\square$

**Lemma 12.** $\forall x \in \mathcal{X}$, $\forall r > \frac{2\lambda^G}{1+\lambda^G}$, and $\forall p \in \mathbb{C}^n$, $C_{2,x}(p) < \frac{\left((1+\lambda^G)r - 2\lambda^G\right)^2}{4 - 2\lambda^G} \implies C_{1,x}(p) < r$.

*Proof.* By Lemma 10, $C_{2,x}(p) < \frac{\left((1+\lambda^G)r - 2\lambda^G\right)^2}{4 - 2\lambda^G} \implies \|p - p_x^*\|_2 < \frac{(1+\lambda^G)r - 2\lambda^G}{\sqrt{2-\lambda^G}}$. Fix $x \in \mathcal{X}$. Recall that $\beta^G(p) = \min\{\arg\min_{i \in \mathcal{Y}}\|p - g_i\|_2\}$. We claim

$$
\|p - p_x^*\| < \frac{(1+\lambda^G)r - 2\lambda^G}{\sqrt{2 - \lambda^G}} \implies \max_i \eta_i(x) - \eta_{\beta^G(p)}(x) < r.
$$

Assume $\|p - p_x^*\| < \frac{(1+\lambda^G)r - 2\lambda^G}{\sqrt{2-\lambda^G}}$ and $\max_i \eta_i(x) - \eta_{\beta^G(p)}(x) \geq r$. By Lemma 11, $\left\|p - g_{\beta^G(p)}\right\| > \left\|p - g_{\min\{\arg\max_i \eta_i(x)\}}\right\|$, contradicting the definition of $\beta^G(p)$. Hence, $C_{1,x}(p) = \max_i \eta_i(x) - \eta_{\beta^G(p)}(x) < r$.

$\square$

Now we're ready to prove Theorem 4.

*Proof of Theorem 4.*

$$
\begin{aligned}
\mathcal{R}_{L^G,P}(f) - \mathcal{R}_{L^G,P}^* &= \int_{\mathcal{X}} C_{1,x}(f(x)) \\
&= \int_{x:d(x)<r} C_{1,x}(f(x)) + \int_{x:d(x)\geq r} C_{1,x}(f(x)).
\end{aligned}
$$

We bound each integral individually.

By Lemma 12, $\forall x \in \mathcal{X}$, $\forall r > \frac{2\lambda^G}{1+\lambda^G}$, and $\forall p \in \mathbb{C}^n$,

$$
C_{1,x}(p) \geq r \implies C_{2,x}(p) \geq \frac{\left((1+\lambda^G)r - 2\lambda^G\right)^2}{4 - 2\lambda^G}. \tag{5}
$$

Hence,

$$
\begin{aligned}
C_{1,x}(p) > \frac{2\lambda^G}{1 + \lambda^G} &\implies C_{2,x}(p) \geq \frac{\left((1 + \lambda^G)C_{1,x}(p) - 2\lambda^G\right)^2}{4 - 2\lambda^G} \\
&\implies C_{1,x}(p) \leq \frac{2\lambda^G}{1 + \lambda^G} + \frac{1}{1 + \lambda^G}\sqrt{(4 - 2\lambda^G)C_{2,x}(p)}.
\end{aligned}
$$

Note the last inequality actually holds for all $p \in \mathbb{C}^n$, that is, it holds even when $C_{1,x}(p) \leq \frac{2\lambda^G}{1+\lambda^G}$. Then,

$$\int_{x:d(x)<r} C_{1,x}(f(x))$$

$$\leq \int_{x:d(x)<r} \frac{2\lambda^G}{1+\lambda^G} + \frac{1}{1+\lambda^G}\sqrt{(4-2\lambda^G)C_{2,x}(f(x))}$$

$$= \frac{2\lambda^G}{1+\lambda^G}P_{\mathcal{X}}(d(X)<r) + \frac{\sqrt{4-2\lambda^G}}{1+\lambda^G}\int_{x:d(x)<r}\sqrt{C_{2,x}(f(x))}$$

$$= \frac{2\lambda^G}{1+\lambda^G}P_{\mathcal{X}}(d(X)<r) + \frac{\sqrt{4-2\lambda^G}}{1+\lambda^G}\left\|\mathbb{1}_{d(x)<r}\sqrt{C_{2,x}(f(x))}\right\|_{P_{\mathcal{X}},1}$$

$$\leq \frac{2\lambda^G}{1+\lambda^G}P_{\mathcal{X}}(d(X)<r) + \frac{\sqrt{4-2\lambda^G}}{1+\lambda^G}\left\|\mathbb{1}_{d(x)<r}\right\|_{P_{\mathcal{X}},2}\left\|\sqrt{C_{2,x}(f(x))}\right\|_{P_{\mathcal{X}},2} \tag{6}$$

$$= \frac{2\lambda^G}{1+\lambda^G}P_{\mathcal{X}}(d(X)<r) + \frac{\sqrt{4-2\lambda^G}}{1+\lambda^G}\sqrt{P_{\mathcal{X}}(d(X)<r)\left(\mathcal{R}_{\ell^G,P}(f)-\mathcal{R}^*_{\ell^G,P}\right)}.$$

In inequality (6), we apply Holder's inequality.

When $C_{1,x}(p) > 0$, $C_{1,x}(p) = \max_i \eta_i(x) - \eta_{\beta^G(p)}(x) \geq \max_i \eta_i(x) - \max_{i\notin\arg\max_j \eta_j(x)}\eta_i(x) = d(x)$. By (5), if $d(x) \geq r$ and $C_{1,x}(p) > 0$, then $C_{2,x}(p) \geq \frac{((1+\lambda^G)r-2\lambda^G)^2}{4-2\lambda^G}$. As $C_{1,x}(p) \in [0,1]$, $d(x) \geq r$ and $C_{1,x}(p) > 0$ $\implies C_{2,x}(p) \geq \frac{((1+\lambda^G)r-2\lambda^G)^2}{4-2\lambda^G}C_{1,x}(p)$. It is trivial that $C_{2,x}(p) \geq \frac{((1+\lambda^G)r-2\lambda^G)^2}{4-2\lambda^G}C_{1,x}(p)$ also holds when $C_{1,x}(p) = 0$. Thus, $\forall x \in \mathcal{X}$ and $\forall p \in \mathbb{C}^n$, $d(x) \geq r \implies C_{2,x}(p) \geq \frac{((1+\lambda^G)r-2\lambda^G)^2}{4-2\lambda^G}C_{1,x}(p) \implies C_{1,x}(p) \leq \frac{4-2\lambda^G}{((1+\lambda^G)r-2\lambda^G)^2}C_{2,x}(p)$. Therefore,

$$\int_{x:d(x)\geq r} C_{1,x}(f(x)) \leq \int_{x:d(x)\geq r} \frac{4-2\lambda^G}{\left((1+\lambda^G)r-2\lambda^G\right)^2}C_{2,x}(f(x))$$

$$\leq \frac{4-2\lambda^G}{\left((1+\lambda^G)r-2\lambda^G\right)^2}\int_{\mathcal{X}} C_{2,x}(f(x))$$

$$= \frac{4-2\lambda^G}{\left((1+\lambda^G)r-2\lambda^G\right)^2}\left(\mathcal{R}_{\ell^G,P}(f)-\mathcal{R}^*_{\ell^G,P}\right).$$

$\square$

# B  Experiment details

In this section, we provide the details of our experiments.

## B.1  Neural networks

In this section, we provide details on the architectures and hyperparameter choices for the neural networks used in our experiments. The architectures and hyperparameters are selected by trial-and-error on a heldout dataset.

### B.1.1  LSHTC1

The proposed embedding strategy adopts a 2-layer neural network architecture, employing a hidden layer of 4096 neurons with ReLU activation. The output of the neural network is normalized to have a Euclidean norm of 1. An Adamax optimizer with a learning rate of 0.001 is utilized together with a batch size of 128 for training. The model is trained for a total of 5 epochs. In order to effectively manage the learning rate, a scheduler is deployed, which scales down the learning rate by a factor of 0.1 at the second epoch.

Our cross-entropy baseline retains a similar network architecture to the embedding strategy, with an adjustment in the output layer to reflect the number of classes. Here, the learning rate is 0.01 and the batch size is 128 for training. The model undergoes training for a total of 5 epochs, with a scheduler set to decrease the learning rate after the third epoch.

Finally, the squared loss baseline follows the architecture of our cross-entropy baseline, with the learning rate and batch size mirroring the parameters of the embedding strategy. As with the embedding strategy, the output is normalized. The model is trained for a total of 5 epochs, and the learning rate is scheduled to decrease after the third epoch.

### B.1.2   DMOZ

For the DMOZ dataset, our proposed label embedding strategy employed a 2-layer neural network with a hidden layer of 2500 neurons activated by the ReLU function. The output of the network is normalized to have a norm of 1. We trained the model using the Adamax optimizer with a learning rate of 0.001 and a batch size of 256. The model was trained for 5 epochs, and the learning rate was scheduled to decrease at the second and fourth epochs by a factor of 0.1.

For the cross-entropy loss baseline, we used the same network architecture with an adjustment in the output layer to match the number of classes. The learning rate was 0.01 and the batch size was 256. The model underwent training for a total of 5 epochs, with the learning rate decreasing after the third epoch as determined by the scheduler.

Lastly, the squared loss baseline utilized the same architecture, learning rate, and batch size as the proposed label embedding strategy. Similarly, the model's output was normalized. The model was trained for 5 epochs, with the learning rate scheduled to decrease after the third epoch.

### B.1.3   ODP

For the ODP dataset, the experiments utilized a neural network model composed of 4 layers. The size of the hidden layers progressively increased from $2^{10}$ to $2^{14}$, then decreased to $2^{13}$. Each of these layers employed the ReLU activation function and was followed by batch normalization to promote faster, more stable convergence. The final layer output size corresponded with the embedding dimension for the label embedding strategy and the number of classes for the cross-entropy and squared loss baselines.

In the label embedding framework, the output was normalized to yield a norm of 1. This model was trained using the Adamax optimizer, a learning rate of 0.001, and a batch size of 2048. The training spanned 20 epochs, with a learning rate decrease scheduled after the 10th epoch by a factor of 0.1.

For the cross-entropy loss baseline, the same network architecture was preserved, with an adjustment to the penultimate layer, reduced by half, and the final output layer resized to match the number of classes. This slight modification in the penultimate layer was necessary to accommodate the models within the 48GB GPU memory. Notably, the neural network output was normalized by dividing each output vector by its Euclidean norm before applying the softmax function, a non-standard operation that significantly enhanced performance. This model was trained using a learning rate of 0.01 over 20 epochs, following a similar learning rate schedule.

Finally, the squared loss baseline used the same architecture as the cross-entropy baseline and the same learning rate and batch size as the label embedding model. Here, the output was also normalized. The model underwent training for 20 epochs, with a learning rate decrease scheduled after the 10th epoch.

### B.2   Elastic net

We aim to solve

$$\min_{W \in \mathbb{R}^{D \times n}} \|XW - Y\|_{fro}^2 + \lambda_1 \|W\|_{1,1} + \lambda_2 \|W\|_{fro}^2, \tag{7}$$

where $X \in \mathbb{R}^{N \times D}$ is the data matrix, the rows of $Y \in \mathbb{R}^{N \times n}$ are embedding vectors.

The problem (7) can be broken down into $n$ independent real-output regression problems of the form

$$\min_{W_j \in \mathbb{R}^D} \|XW_j - Y_j\|_{fro}^2 + \lambda_1 \|W_j\|_1 + \lambda_2 \|W_j\|_2^2,$$

where $W_j$ is the $j$-th column of $W$ and $Y_j$ is the $j$-th column of $Y$. Consequently, We can distribute the $n$ real-output regression problems across multiple cores.

We develop a regression variant of the Fully-Corrective Block-Coordinate Frank-Wolfe (FC-BCFW) algorithm (Yen et al., 2016) and use it to solve the real-output regression problems. As the solver operates iteratively, we set it to run for a predefined number of iterations, denoted as $N_{\text{iter}}$. The chosen hyperparameters are outlined in table 5.

| Dataset | $\lambda_1$ | $\lambda_2$ | $N_{\text{iter}}$ |
|---------|------|------|------|
| LSHTC1 | 0.1 | 1 | 20 |
| DMOZ | 0.01 | 0.01 | 20 |
| ODP | 0.01 | 0.1 | 40 |

Table 5: Hyperparameters for elastic net.

### B.3 Practical issues

### B.3.1 Choice of embedding dimension

In practical settings, the choice of the embedding dimension $n$ depends on available computational resources. Under a fixed computational budget, our theory recommends opting for an embedding with minimal coherence. When the type of embedding is fixed but computational resources are flexible, we advise treating the embedding dimension $n$ as a tunable parameter. Specifically, it is beneficial to incrementally increase $n$, perhaps exponentially, and observe the performance. This process should continue until the improvement in performance plateaus or the additional computational cost becomes prohibitively high.

### B.3.2 Types of embedding matrices

In our experiments, we used four types of embeddings. Random embeddings stand out for their simplicity and flexibility. They can be generated with a single line of code, and the embedding dimension can be any positive integer. However, their drawbacks include the need for explicit storage and potentially higher coherence compared to some deterministic matrices. On the other hand, deterministic matrices like those constructed using Nelson's method can achieve lower coherence, generally leading to better performance. They do not require explicit storage, as individual columns can be generated on demand. The trade-off is that they are more complex to implement, and the options for embedding dimensions are more limited (*e.g.*, Nelson's construction requires prime numbers for the embedding dimension).

### B.4 Used assets

We list the existing code used in our experiments.

- PD-Sparse (Yen et al., 2016): `https://github.com/a061105/ExtremeMulticlass` (BSD-3-Clause license).

- PPD-Sparse (Yen et al., 2017a): `https://github.com/a061105/AsyncPDSparse`.

- Parabel (Prabhu et al., 2018): `http://manikvarma.org/code/Parabel/download.html`.

- AnnexML (Tagami, 2017): `https://github.com/yahoojapan/AnnexML?tab=Apache-2.0-1-ov-file`(Apache-2.0 license).

- WLSTS(Evron et al., 2018): `https://github.com/ievron/wltls/?tab=MIT-1-ov-file`(MIT License).

## B.5 Visualization of Embeddings

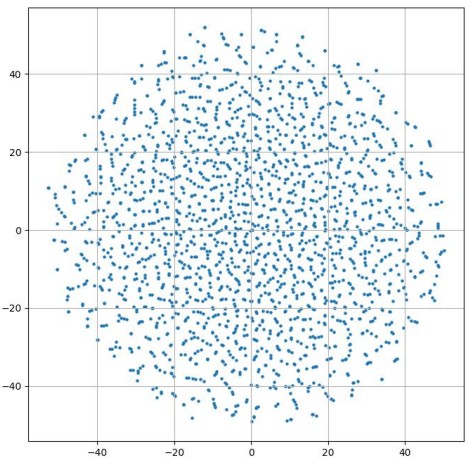

(a) t-SNE plots of 2,048 Rademacher embedding vectors in $\mathbb{R}^{128}$.

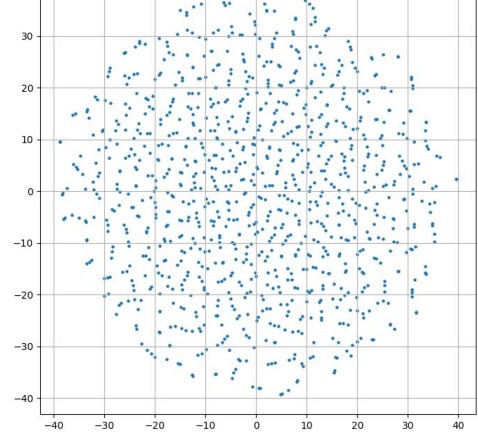

(b) t-SNE plots of 2,048 Gaussian embedding vectors in $\mathbb{R}^{128}$.

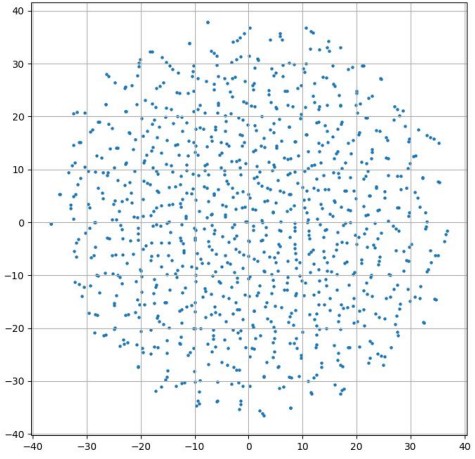

(c) t-SNE plots of 2,048 complex Guassian embedding vectors in $\mathbb{R}^{128}$.

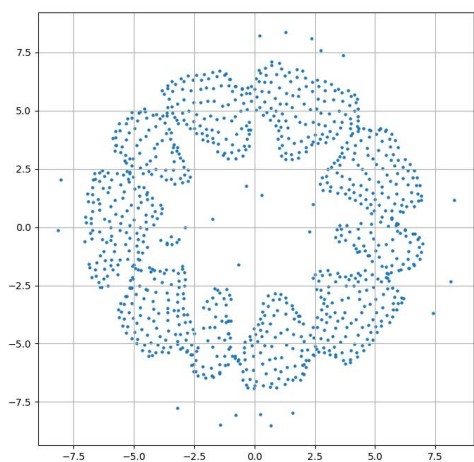

(d) t-SNE plots of 2,048 Nelson's embedding vectors in $\mathbb{R}^{127}$.

Figure 2: t-SNE plots illustrating the distribution of vectors from four embedding types.

To better understand the structural differences among various embedding types, we visualize 2,048 embedding vectors from each method using t-SNE, as shown in Figure 2. The embeddings include Rademacher, Gaussian, complex Gaussian, and Nelson's construction. The random embeddings all exhibit roughly isotropic spreads in the low-dimensional projection. In contrast, the Nelson embeddings display a more organized and geometrically constrained distribution.

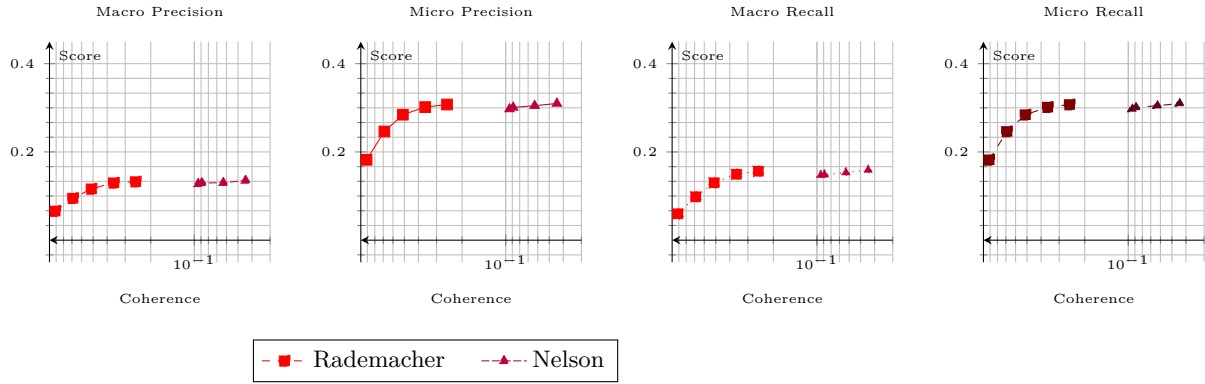

Figure 3: Coherence vs metric score on LSHTC1 comparing Rademacher and Nelson embeddings across precision and recall (macro and micro).

### B.6 Precision and recall

For a set of $C$ classes, let $\mathrm{TP}_c, \mathrm{FP}_c, \mathrm{FN}_c$ denote true positives, false positives, and false negatives for class $c$. Then

$$\mathrm{Precision}_c = \frac{\mathrm{TP}_c}{\mathrm{TP}_c + \mathrm{FP}_c}, \quad \mathrm{Recall}_c = \frac{\mathrm{TP}_c}{\mathrm{TP}_c + \mathrm{FN}_c}.$$

Macro-averages compute the unweighted mean across classes,

$$\mathrm{Macro\text{-}Precision} = \frac{1}{C}\sum_{c=1}^{C} \mathrm{Precision}_c, \quad \mathrm{Macro\text{-}Recall} = \frac{1}{C}\sum_{c=1}^{C} \mathrm{Recall}_c,$$

while micro-averages pool counts across classes before computing,

$$\mathrm{Micro\text{-}Precision} = \frac{\sum_{c=1}^{C} \mathrm{TP}_c}{\sum_{c=1}^{C}(\mathrm{TP}_c + \mathrm{FP}_c)}, \quad \mathrm{Micro\text{-}Recall} = \frac{\sum_{c=1}^{C} \mathrm{TP}_c}{\sum_{c=1}^{C}(\mathrm{TP}_c + \mathrm{FN}_c)}.$$

To assess whether the benefits of low coherence extend beyond classification accuracy, we include additional plots showing four other performance metrics: macro-precision, macro-recall, micro-precision, and micro-recall. Figure 3 shows how these metrics vary with the coherence of the embedding matrix for LSHTC1 using Rademacher and Nelson constructions. Consistent with the accuracy trends reported in the main paper, we observe a clear inverse relationship between coherence and all measures. As coherence decreases, moving rightward along the reversed x-axis, all four scores consistently improve.

## C  Extended discussion

In this section, we provide supplementary insights and theoretical context that extend the main exposition.

### C.1  Consistent loss functions

Let $(X, Y) \sim P$ with $Y \in \{1, \ldots, K\}$ and a function $f : \mathcal{X} \to \mathbb{R}^K$. For a surrogate $\phi : \mathbb{R}^K \times \{1, \ldots, K\} \to \mathbb{R}$ define the risk $R_\phi(f) = \mathbb{E}\big[\phi\big(f(X), Y\big)\big]$. We say $\phi$ is *(Fisher) consistent* w.r.t. the 0–1 loss $\ell$ if every minimiser $f^\star \in \arg\min_f R_\phi(f)$ induces a Bayes–optimal classifier $g^\star(x) = \arg\max_k f_k^\star(x)$, i.e. $\inf_f R_\phi(f) = R_\phi(f^\star) \Rightarrow \inf_g R_\ell(g) = R_\ell(g^\star)$ (Zhang, 2004; Tewari and Bartlett, 2007; Steinwart, 2007). Canonical consistent surrogates include the mean-squared error $\phi_{\mathrm{MSE}}(f, y) = \|e_y - f\|_2^2$ and the multinomial logistic (cross-entropy) loss $\phi_{\log}(f, y) = -\log\frac{\exp(f_y)}{\sum_k \exp(f_k)}$. In contrast, margin-based hinge variants such as $\phi_{\mathrm{hinge}}(f, y) = \max_{j \neq y}\big\{1 + f_j - f_y\big\}_+$ are known to be *inconsistent* in the multiclass setting (Tewari and Bartlett, 2005).

### C.2 Nelson's construction

Let $r \geq 2$ be a natural number and $p > r$ be a prime number. Define an index list $a := (a_1, \ldots, a_r)$ where the $a_i$'s are integers in $[1, n]$. For each $a$, we define

$$F(a, u) := \sum_{j=1}^{r} a_j u^j.$$

The $n$ dimension embedding is constructed as

$$g^a := \frac{1}{\sqrt{n}} \left( \exp\left( \frac{2\pi i}{n} F(a, 1) \right), \ldots, \exp\left( \frac{2\pi i}{n} F(a, n) \right) \right)^T.$$

Given $n$ and $r$, each embedding vector is uniquely determined by $a$, yielding an embedding capacity of $n^r$. The magnitude of the inner product between $g^a$ and $g^{a'}$ is bounded by $\frac{r-1}{n}$ when $a \neq a'$ (Nelson and Temlyakov, 2011).

### C.3 Dynamically growing label space

In scenarios with dynamically growing label spaces, random embedding matrices present no significant challenges: as new labels arrive, we can simply sample their embeddings independently and identically distributed (i.i.d.). For deterministic constructions, the capacity depends on the construction algorithm of the embedding matrices. In Nelson's framework, the number of available low-coherence embeddings scales as $n^r$, allowing expansion to a polynomially large number of labels while maintaining desirable geometric properties. However, once the $(n^r - 1)$ cap is reached, the labels have to be 'rehashed'.

