# OpenReview forum: "Label Embedding via Low-Coherence Matrices"
_TMLR — Accepted by TMLR_

### Review · Reviewer_d4j8 · 2025-04-13

**Summary Of Contributions:**

This paper focuses on label embedding, which is a framework for the multi-class classification problem, where each label is represented by a distinct vector. Instead of doing classification, the target is now not outputting the classification score directly, but regressing the label embedding so that it matches the correct label. This paper further focuses on the theoretical foundation of the label embedding setting in the extreme multiclass classification setting (the number of classes is very large). To this end, this paper presents an excess risk bound that shows the trade-off between computation and statistical efficiency. The core part of such a trade-off is the coherence of the embedding matrix, one can increase the embedding dimension to reduce such coherence but sacrifice the computational efficiency. Based on the theoretical findings, the author conduct experiments with a simple, scalable algorithm. They conduct experiments on several datasets to show the superiority of such an algorithm.

**Audience:**

Yes

**Claims And Evidence:**

Yes

**Requested Changes:**

So the previous section.

**Strengths And Weaknesses:**

Strengths

The paper is well organized, and the problem setting is easy to follow. The theory is complemented by sufficient explanation and mathematical steps. The conclusion is intuitive and easy to follow. Both the excess risk bound of the general case and the improvement under low noise are well displayed clearly. The author compared the LOCOLE (LOw COherence Label Embeding) method with several state-of-the-art methods under the Meta-Algorithm 1 on several datasets and achieved significant performance improvement.


Weaknesses
According to the author, Meta-Algorithm 1 is not part of the contribution. Besides, the way to construct the low-coherence matrix. It is my understanding that the main contribution is the theoretical analysis to show the relationship between the excess risk and the embedding matrix coherence. If this is the case, can the author make such a part clearer in the introduction?

The author mainly conducts the experiments under the setting that matrix G is fixed. However, the author also claims that the proposed analysis also applies to the case where G is trained jointly with the f as well. I am wondering for the cases where G is trained jointly with the f, what would G look like, do they have a similar low-coherence property, or this is dataset dependent? From my understanding, deep neural networks are good at capturing some inherence relationships within the dataset.  For example, in some cases for image classification, some classes are more similar than others by definition (for example, dog vs cat, compared to dog vs grass) and show have higher coherence, under such cases, I am not sure whether imposing the low-coherence as a strong prior has some other implications.

---

> ### Author Response · Authors · 2025-04-27
>
> Thank you for the thoughtful feedback. We appreciate your careful reading and constructive suggestions. Please see our responses below:
>
> 1. Clarification in the Introduction
> Thank you for the suggestion. We agree that this point should be made clearer. In our next revision, we will explicitly state in the introduction that we do not propose a new compressed sensing matrix, and that Meta-Algorithm 1 serves as a summary of existing algorithms.
>
> 2. Joint Training of G and f
> Thank you for raising this point. Our excess risk bound holds independently of how G is obtained. While joint training of G and f is indeed possible, we did not explore it empirically because there is currently no standard or widely accepted methodology for doing so. Although heuristic approaches exist, we are concerned that they may introduce uncontrolled confounding factors, complicating empirical interpretation. We will add a discussion of this consideration to the paper to clarify our decision.

---

### Review · Reviewer_4dpB · 2025-04-19

**Summary Of Contributions:**

This paper studies classification error under the dimensionality reduction of label embedding, with mean squared error (MSE) as the surrogate loss. This setting is particularly relevant for extreme classification, where the number of classes is large. An upper bound on the excess risk is provided in Theorem 4, which constitutes the paper's main contribution. The authors also propose an intuitive regression-based algorithm for classification under label embedding.

**Audience:**

Yes

**Broader Impact Concerns:**

This paper is primarily theoretical and does not raise immediate societal or ethical concerns. However, given the relevance of label embeddings to large-scale classification tasks (e.g., in recommendation systems or language models), it is important to remain mindful of potential biases that can be introduced or amplified through embedding representations, particularly in applications involving sensitive or imbalanced label distributions.

**Claims And Evidence:**

Yes

**Requested Changes:**

**To address the weaknesses outlined above, I suggest the following specific changes:**

- Consolidate and clearly explain all notations in a single location, rather than introducing them gradually before Section 3.1 and again at the beginning of Section 3.2.
- Define the notion of consistency and provide concrete examples of consistent and inconsistent surrogate losses.
- Include a simple numerical example to illustrate how the embedding matrix is constructed and how class labels are mapped into the embedding space.
- Add a clarifying note to Proposition 2 explaining why, despite preserving expressiveness, a low embedding dimensionality combined with a convex surrogate loss results in irreducible excess risk.
- Revise the experimental section to improve clarity and completeness. In particular, clarify whether Table 3 reports only LOCOLE results, explain the behavior of non-LOCOLE baselines (e.g., why they appear as horizontal lines), and ensure that all tables and figures are sufficiently captioned and self-contained.

**Strengths And Weaknesses:**

### **Strengths**

- Classification with label embedding is an important problem in extreme classification and natural language processing.

- The upper bound on the excess risk derived in this paper provides theoretical insights that can potentially guide the design of surrogate loss functions, embedding strategies, and learning algorithms.

### **Weaknesses**
- The paper's organization and presentation can be significantly improved; it is not easily accessible, even for a technically familiar audience. For example, the introduction of notations is scattered, the definition of matrix coherence is confusing, and it lacks proper motivation and interpretation. Additionally, the concept of "consistency" is not defined, and no examples of consistent or inconsistent loss functions are provided.

- Proposition 2 can be misinterpreted. While it is correct that no expressiveness is lost by representing a classifier as a composition of a decoder and a measurable function, choosing an embedding dimensionality \( n < C - 1 \) under a convex surrogate loss introduces irreducible error, as discussed in Section 2.2. This subtle but important point should be clarified.

- The experimental section is poorly presented. For instance, it is unclear whether Table 3 reports results only for LOCOLE. Summarizing the results of Table 3 in just one or two sentences is insufficient. Moreover, the reason why non-LOCOLE methods are plotted as horizontal lines is not explained. Are these methods unaffected by embedding dimensionality or coherence? This is a critical point, and the authors have sufficient space to elaborate on it. Finally, the tables and figures are minimally captioned and not self-contained.

---

> ### Author Response · Authors · 2025-05-11
>
> Thank you for the detailed and thoughtful feedback. We appreciate the reviewer’s recognition of our theoretical contributions and the relevance of our work to extreme classification. We will address the concerns raised with the following improvements in the next revision:
>
> 1. Notation Presentation: We will reorganize the presentation of notations to ensure they are introduced clearly and consolidated in one place before being used in the main results.
>
> 2. Consistency of Surrogate Losses: We will define the notion of consistency formally and cite standard references. We will also include concrete examples of consistent (e.g., MSE, logistic) and inconsistent (e.g., certain hinge losses) surrogate loss functions.
>
> 3. Embedding Matrix Construction: The constructions of the Rademacher, Gaussian, and C-Gaussian embedding matrices are already specified in the experimental section as entrywise i.i.d. sampling procedures. We will include a description of Nelson's algorithm in the appendix for completeness and clarity.
>
> 4. Clarification of Proposition 2: We will add a clarifying remark to Proposition 2, emphasizing that while function composition preserves expressiveness, choosing a low-dimensional embedding in conjunction with a convex surrogate can introduce irreducible error. This distinction will be highlighted to avoid misinterpretation.
>
> 5. Experimental Section Improvements: Table 3 reports both LOCOLE and competitor results; the competitor acronyms are listed in Section 4.1. We will clarify this in the table caption and in the paper. Regarding the plots, non-LOCOLE baselines appear as horizontal lines because they do not depend on the embedding dimensionality—this will be made explicit in the figure captions. We will also expand the accompanying descriptions to better contextualize the findings and ensure all tables and figures are self-contained.

---

### Review · Reviewer_FA3q · 2025-06-10

**Summary Of Contributions:**

The current manuscript provides an analysis of label embedding in the context of extreme multiclass classification, where the number of labels can be very large. Label embedding replaces one-hot label vectors with dense vector representations, enabling scalable learning. To this end, the authors introduce  LOCOLE (Low COherence Label Embedding) that: i) derives an excess risk bound for label embedding using low-coherence matrices. ii)demonstrates that coherence (the maximum correlation between label vectors) directly influences classification performance. iii) they prove under the Massart noise condition that the statistical penalty of embedding vanishes if coherence is sufficiently low. iv) they also present LOCOLE, and empirically validate its performance across large-scale datasets such as LSHTC1, DMOZ, and ODP. Experimental results confirm the theory: classification accuracy improves as matrix coherence decreases.

**Audience:**

Yes

**Claims And Evidence:**

Yes

**Requested Changes:**

I have a few requests from the authors and would like to consider them, or at least provide discussion on these topics:

### Suggestions on theoretical perspective:
- can you please provide theoretical insight/discussion into extending the method to multi-label classification, a natural generalization?
- Can you please  consider relax assumption and discuss more realistic noise models beyond the Massart condition or test its validity empirically?
-

### Suggestions on improving the  experimental setup
- can you please incorporate learned embeddings (e.g., via deep co-training) to demonstrate generalizability beyond fixed low-coherence matrices?
- can you please study how your proposed method handles class imbalance, noisy labels, or dynamically growing label spaces?
- can you please provide deeper insight into how coherence relates to other performance metrics (e.g., precision, recall) and not just accuracy?
- If it is possible, can you please visualize the embedding spaces (e.g., t-SNE plots of embeddings with varying coherence) this could help make abstract ideas more concrete.

**Strengths And Weaknesses:**

### Strengths:
I thhink the current manuscript is relevant to the TMLR community since:
- the paper presents a novel excess risk bound that links label embedding accuracy to matrix coherence.
- It delivers rigorous mathematical backing to justify why low-coherence embeddings are useful.
- and it shows that under Massart noise conditions, classification using label embedding becomes statistically consistent.
- Their proposed Meta-Algorithm 1 is simple and modular, making it straightforward to plug into different model architectures and embedding strategies.
- They conducted several experimental results (i.e, on three large-scale datasets) and some ablation analysis and validated that LOCOLE is efficient for the task of multiclass classification.

### Weaknesses

- One of the weaknesses of this study is the *fixed size of embedding*; besides suggesting generality, the experiments are restricted to fixed embeddings. This leaves out learned embeddings or embeddings using side-information, which are prevalent in practice.
- Another one is the *Massart noise condition* that is a strong assumption and may not hold in real-world datasets, while it plays a critical role in justifying the *lossless embedding claims*.
- The paper focuses solely on multiclass classification. While the authors mention extensions to multilabel or zero-shot settings but they are not demonstrated.
-there is little discussion on practical limitations, such as handling imbalanced labels or noise in embeddings.

---

> ### Author Response · Authors · 2025-06-26
>
> We thank the reviewer for the thoughtful and constructive feedback. Below we address each concern raised in the review:
>
> 1. Theoretical Extension to Multi-Label Classification
>
> We agree that extending our framework to multi-label classification is a natural next step. We have already developed preliminary theoretical results for the multi-label setting. However, incorporating them would have substantially expanded the scope of the current paper. We will probably explore this direction in future work and will add a forward-looking discussion in the revised version.
>
> 2. Massart Noise Assumption
>
> We respectfully argue that the Massart noise model is not overly restrictive. It is widely accepted as a meaningful condition in theoretical learning literature. For example, Chandrasekaran et al. (2024) write that “it is reasonable to consider Massart noise to be a more realistic model of real-life noise” [1], while Diakonikolas et al. (2021) note that "the Massart model is a natural semi-random input model"  and “algorithms that learn in the presence of Massart noise are likely to be less brittle” [2].
>
> Like many distributional assumptions (e.g., i.i.d. sampling), Massart noise cannot generally be verified on real-world datasets, except for synthetic ones.
>
> 3. Generalizability Beyond Fixed Low-Coherence Embeddings
>
> There is currently no standardized method for label embedding across general classification tasks, and learned embeddings can introduce confounding factors. Our design choice—to focus on fixed embeddings—was intentional to isolate the impact of coherence and to provide clean empirical validation of our theoretical results. We will add this clarification to the paper.
>
> 4. Handling Class Imbalance, Noisy Labels, and Growing Label Spaces
>
> Each of these directions is important and substantial. Our current datasets already exhibit significant class imbalance due to their long-tailed label distributions. Label noise, while a separate challenge, could potentially be mitigated by recent simple methods like “Label Noise: Ignorance Is Bliss” (Zhu et al., 2024) [3], which uses frozen foundation models followed by a linear head—an approach naturally compatible with extreme classification.
>
> For dynamically growing label spaces, random embedding matrices pose no difficulty: we can sample new embeddings i.i.d. as new labels arrive. Deterministic constructions such as Nelson’s allow up to n^r low-coherence embeddings, so labels can be expanded within that capacity. We are happy to add this discussion in the next version.
>
> 5. Coherence vs. Other Metrics (Precision/Recall)
>
> We agree with the reviewer that coherence may affect a variety of metrics beyond accuracy. We will include additional experimental results (e.g., precision, recall, and F1) in the appendix to provide a broader perspective on the benefits of low coherence.
>
> 6. Embedding Space Visualizations
>
> We appreciate the suggestion and will include t-SNE visualizations of embeddings with varying coherence to make our findings more intuitive.
>
> References
>
> [1] Chandrasekaran, V., Menon, A. K., Song, L., & Williamson, R. C. (2024). Learning Noisy Halfspaces with a Margin: Massart is No Harder than Random. arXiv:2402.04173.
>
> [2] Diakonikolas, I., Kane, D. M., Stewart, A., & Sun, X. (2021). Boosting in the Presence of Massart Noise. In Conference on Learning Theory (COLT).
>
> [3] Zhu, J., Zhai, X., Bengio, S., Kolesnikov, A., & Houlsby, N. (2024). Label Noise: Ignorance is Bliss. arXiv:2402.00758.

---

### Decision · Action_Editor_KPd4 · 2025-07-30

**Recommendation:** Accept with minor revision

**Additional Comments:**

During the review process, the reviewers found the claims to be well-supported through rigorous mathematical derivations and experiments that clearly showed improvements in performance. The reviewers also recognized the problem setting as important and saw clear potential for the theoretical analysis to lead to improved label embedding algorithms in the future.

Opinions were mixed on clarity, with concerns being raised about the presentation of the experimental results. There were also questions about the restrictiveness of some of the assumptions, including the fixed nature of the embedding and the Massart noise condition.  In addition, the novelty of the work was found to be somewhat limited.

The criteria for acceptance to TMLR are (1) sufficient evidence to support claims and (2) interest to the community. In particular, novelty is not a stated criterion and thus any potential concerns about novelty do not factor into the decision. Theoretical assumptions are explicitly recognized as limitations in the paper and therefore the claims are appropriately scoped. As both of the TMLR criteria are satisfied, I recommend acceptance.

Nevertheless, there are outstanding issues regarding the presentation of the experimental results as outlined by Reviewer 4dpB. The authors are strongly encouraged to revise the manuscript accordingly, by expanding the discussion of the results and ensuring that figure and table captions are self-contained.

**Audience:**

Yes

**Audience Explanation:**

Researchers working on extreme classification, zero-shot classification, and theoretical foundations of machine learning would likely be interested in the contributions of this work.

**Claims And Evidence:**

Yes

**Claims Explanation:**

This paper provides an analysis of _label embedding_, a technique to handle classification with a large number of classes. In label embedding, classification is treated as regression into an embedding space populated with label vectors. In this work, the classification error is shown to be related to the coherence of the label embeddings: decreasing the coherence (e.g. by increasing embedding dimensionality) leads to a corresponding improvement in classification accuracy.

Key contributions of the work include:
1. A theoretical bound that quantifies the relationship between coherence and excess risk.
2. A characterization of excess risk in the Massart noise setting.
3. A label embedding algorithm called LOCOLE, motivated by the theoretical analysis, that is effective for large-scale classification problems.

The claims of the paper are supported by theory (contributions 1 and 2) and experimental results on multiple datasets (contribution 3).